# Digital Interventions for the Rehabilitation of First-Episode Psychosis: An Integrated Perspective

**DOI:** 10.3390/brainsci15010080

**Published:** 2025-01-16

**Authors:** Annarita Vignapiano, Francesco Monaco, Ernesta Panarello, Stefania Landi, Benedetta Di Gruttola, Raffaele Malvone, Vassilis Martiadis, Fabiola Raffone, Alessandra Marenna, Maria Pontillo, Valeria Di Stefano, Martina D’Angelo, Luca Steardo, Giulio Corrivetti

**Affiliations:** 1Department of Mental Health, ASL Salerno, 84125 Salerno, Italy; a.vignapiano@aslsalerno.it (A.V.); f.monaco@aslsalerno.it (F.M.); s.landi@aslsalerno.it (S.L.); bennydgr2@gmail.com (B.D.G.); raffmalvone@gmail.com (R.M.); corrivetti@gmail.com (G.C.); 2European Biomedical Research Institute of Salerno (EBRIS), 84125 Salerno, Italy; a.marenna@ebris.eu; 3Department of Mental Health, ASL Napoli 1 Centro, 80125 Napoli, Italyfabiolaraffone@gmail.com (F.R.); 4Child & Adolescent Neuropsychiatry Unit, Bambino Gesù Children’s Hospital, IRCCS, 00146 Rome, Italy; maria.pontillo@opbg.net; 5Psychiatric Unit, Department of Health Sciences, University Magna Graecia of Catanzaro, 88100 Catanzaro, Italy; valeria.distefano@studenti.unicz.it (V.D.S.); martina.dangelo001@studenti.unicz.it (M.D.); steardo@unicz.it (L.S.J.)

**Keywords:** digital device, first-episode psychosis, mobile mental health apps, mental health technologies

## Abstract

Background: The integration of digital health technologies has transformed mental healthcare, particularly for young adults with First-Episode Psychosis (FEP). Digital interventions, such as telepsychiatry and mobile applications, address barriers like social stigma, restricted access to services, and the urgency of timely care. Methods: A systematic literature review was conducted using PubMed and APA PsycINFO. Included studies focused on randomized controlled trials, cohort studies, and open studies of digital interventions for FEP rehabilitation. Exclusion criteria included meta-analyses, reviews, and irrelevant studies. Results: Smartphone applications like Horyzons-Canada facilitated peer support and symptom management, while tools such as My Journey 3 emphasized the need for better user engagement. Digital innovations, including Cognitive Behavioral Therapy for insomnia (CBT-I) and telepsychiatry, demonstrated improvements in recovery and treatment adherence. Discussion: Digital interventions significantly enhance the accessibility and quality of FEP care, supporting functional recovery and patient engagement. Future research should prioritize longitudinal studies, user engagement strategies, and advanced analytics to develop personalized, scalable solutions.

## 1. Introduction

### 1.1. The Impact of Digital Technologies on the Management of First-Episode Psychosis (FEP)

The rapid advancement of digital technologies has brought about a transformative shift in the field of mental healthcare, particularly benefiting young adults navigating the challenges of first-episode psychosis (FEP).

These innovations have revolutionized the way care is delivered, breaking down barriers such as geographic limitations, stigma, and resource shortages [1]. Through telepsychiatry, mobile health applications, and virtual support platforms, young individuals now have increased access to timely, personalized, and evidence-based interventions [2]. Furthermore, digital tools enhance engagement by meeting patients in the digital spaces they are familiar with, fostering a sense of comfort and accessibility [3]. This transformation empowers patients and equips healthcare providers with valuable insights through real-time data, enabling more precise and proactive management of FEP [4]. Delayed treatment of psychosis may cause an increase in the likelihood of aggressive behaviors, suicidal ideation, and interactions with law enforcement. During the first few months of a psychotic disorder, timely diagnosis is crucial to prevent negative impacts on the patient’s personal and social functioning [5]. However, in clinical practice, significant challenges persist in diagnosing, engaging, and initiating treatment for individuals experiencing psychosis.

### 1.2. Digital Rehabilitation for First-Episode Psychosis: Objectives and Scope of the Review

Despite widespread recognition of the critical importance of early intervention in improving outcomes, delays in care remain a pressing issue. This review focuses specifically on rehabilitation-oriented digital interventions, addressing their potential to improve recovery, functioning, and quality of life in individuals with FEP. While other digital interventions exist for early detection or symptom monitoring, the rehabilitation phase remains critical for achieving long-term recovery, requiring tailored, technology-driven approaches. By narrowing the scope to rehabilitation, this review aims to synthesize evidence on the effectiveness of digital tools in supporting functional recovery and enhancing treatment outcomes. The average duration of untreated psychosis (DUP) often extends from 1 to 3 years, a concerning statistic given the association between prolonged DUP and worse clinical, functional, and social outcomes [6]. Several factors contribute to these delays, including stigma surrounding mental illness, limited access to specialized services, and a lack of awareness among patients, families, and healthcare providers regarding the early signs of psychosis [7]. In some cases, symptoms may be dismissed or misattributed to other conditions, further postponing appropriate care. Geographic disparities and inadequate availability of specialized early intervention services exacerbate these delays, particularly in underserved or rural areas [8]. Addressing these barriers requires multifaceted strategies, including public education campaigns to reduce stigma and improve mental health literacy, training programs to equip primary care providers with tools for early identification, and expanded access to specialized early intervention teams. Integrating digital technologies, such as telepsychiatry and mobile health tools, also holds promise for reducing delays by improving access to timely assessments and support, especially in areas with limited mental health resources [1]. This prolonged delay highlights the urgent need for more effective strategies to identify and manage individuals with psychosis, including improved access to mental health services [8].

A promising approach to address this persistent issue is through the use of digital tools, such as mobile applications and online platforms, to facilitate early symptom monitoring, improve access to care, and bring about more immediate communication between patients and healthcare professionals. Recent research highlights that telepsychiatry, smartphone applications, and various digital mental health interventions are not only viable alternatives to traditional face-to-face consultations, but are also capable of significantly enhancing patient engagement, facilitating recovery, and optimizing treatment outcomes [9]. While digital interventions encompass diverse aspects of mental healthcare, including early detection and symptom monitoring, this review specifically focuses on their role in rehabilitation. Digital tools in rehabilitation aim to improve functional recovery, promote adherence to treatment, and facilitate engagement with therapeutic processes. By narrowing the scope to rehabilitation, this review seeks to evaluate the effectiveness of these tools in supporting individuals recovering from FEP, a critical phase for achieving long-term functional outcomes. Telepsychiatry has emerged as a particularly valuable resource, enabling mental health professionals to connect with patients remotely through secure video conferencing platforms. This mode of consultation has proven effective in maintaining continuity of care, especially during times when in-person visits are impractical or unsafe, such as during the COVID-19 pandemic [9]. For instance, interventions such as Acceptance and Commitment Therapy combined with ecological momentary intervention [10] highlight the diverse applications of digital tools. However, this review focuses specifically on rehabilitation-oriented interventions, addressing a critical yet underexplored phase of FEP care. By narrowing the scope to rehabilitation, this review aims to provide a comprehensive synthesis of evidence on the role of digital tools in supporting recovery and functional improvement in individuals with FEP.

### 1.3. Challenges and Opportunities in Implementing Digital Technologies for FEP Rehabilitation

The necessity of adhering to physical distancing guidelines has not only prompted a surge in the use of telepsychiatry but has also spurred the development of user-friendly platforms that expand access to care for young adults who may otherwise struggle to seek help. Moreover, smartphone applications specifically designed for mental health support have gained traction by offering users personalized resources, symptom tracking and interactive tools that enhance coping mechanisms [11]. These applications promote self-management, empower patients to take charge of their mental health, and provide a sense of community through peer support features. By integrating these digital solutions into therapeutic frameworks, healthcare providers can promote more robust therapeutic alliances with their patients, improving overall treatment adherence and outcomes. Although digital mental health interventions have demonstrated their effectiveness, it is crucial to recognize the challenges that accompany their implementation. Recent scientific evidence has highlighted the importance of Digital Medicine Systems (DMS) in improving treatment adherence among patients with FEP [12]. Specifically, these studies have demonstrated good acceptance and sustainable use of the Digital Medicine System (DMS) within the general patient population, suggesting that this approach may represent an innovative strategy for objectively monitoring the intake of antipsychotic medications [13,14]. Overall, the findings indicate that DMS could be considered a valuable tool for remote monitoring and therapeutic support for FEP patients, not only enhancing medication adherence but also increasing overall therapeutic compliance. Additionally, it facilitates a smoother integration of digital technologies into the clinical management of psychotic disorders [15]. Despite the promise of these technologies, user engagement remains a significant hurdle. Many individuals, particularly those newly experiencing psychosis, may encounter technical difficulties, lack familiarity with digital platforms, or feel overwhelmed by the sheer volume of available resources. Addressing these barriers through improved onboarding processes, user education, and continuous technical support is essential for maximizing the potential of these interventions [16]. Furthermore, the COVID-19 pandemic has accelerated the need for the ongoing evaluation and adaptation of digital mental health services, particularly as the landscape of healthcare continues to evolve. As we navigate post-pandemic realities, there is an opportunity to assess the long-term impact of these digital tools on recovery trajectories and overall mental health outcomes for young adults with FEP [17]. Researchers and clinicians must collaboratively explore innovative approaches to integrate these technologies effectively into routine care, ensuring that they meet the unique needs of this population. This article synthesizes findings from a range of contemporary studies to critically evaluate the efficacy, challenges, and future directions of digital interventions in the management of FEP. Leveraging digital interventions, clinicians can enhance patient engagement, improve treatment adherence, and reduce barriers to care, such as geographic limitations and stigma. Policymakers can utilize these findings to allocate resources toward scalable and cost-effective digital health technologies, thereby improving access to care for underserved populations.

By examining the intersection of technology, mental health care delivery, and patient experiences, our review aims to inform best practices and promote the development of evidence-based, scalable solutions that can revolutionize the treatment of young adults facing first-episode psychosis. These advancements offer not only the potential to improve individual outcomes but also the opportunity to create a more inclusive and responsive mental healthcare system.

## 2. Methods

### 2.1. Search Strategy and Study Eligibility Criteria

A systematic search was conducted using two major bibliographic databases: PubMed and APA PsycINFO. Articles published between 1 January 2014 and 30 October 2024 were included. We used the following search terms: first episode AND psychosis AND digital AND rehabilitation’ OR ‘first episode AND psychosis AND digital AND diagnosis. Studies were chosen based on these inclusion criteria: randomized controlled trial, retrospective study, cohort study, open study, expert opinion, concerning conceptualization, diagnosis of First-Episode Psychosis (FEP) according to DSM-5 and ICD-10, studies published in English, studies carried out in humans and studies published in journals indexed in Embase or Medline. Manuscripts published in non-English languages were excluded due to resource constraints and the lack of multilingual expertise within the research team. Studies were included if they were explicitly focused on digital tools employed in the rehabilitation of FEP patients, aligning with our objective to explore the role of these interventions in supporting recovery and functional improvement. The exclusion criteria were meta-analysis, review, duplicates, comments, editorials, case reports/case series, theses, proceedings, letters, short surveys and notes, studies irrelevant to the topic, unavailable full-text, and studies that do not meet inclusion criteria. The study selection process was conducted systematically to ensure rigor and transparency, as illustrated in the PRISMA flow diagram, and it was recorded on PROSPERO (registration n° CRD42024626343). A comprehensive search was performed across two major databases: PubMed (n = 20) and APA PsycINFO (n = 23), yielding a total of 43 records. After removing duplicates (n = 2), 41 records remained for screening. During the screening phase, titles and abstracts were assessed for relevance based on predefined inclusion criteria. A total of 29 records were excluded at this stage for the following reasons: 4 were meta-analyses, 5 did not focus on FEP, 13 did not include studies on digital systems, 5 were reviews, and 2 were case reports. Subsequently, 12 full-text articles were assessed for eligibility. Of these, 2 articles were excluded as one did not focus on FEP, and the other did not evaluate digital systems. Studies were excluded primarily for not providing original data (e.g., meta-analyses and reviews), for lacking a focus on FEP, or for not evaluating digital systems, ensuring that the review remained aligned with its specific scope and objectives (Figure 1).

A detailed summary of the included studies, including study design, participant characteristics, and intervention details, is presented in Table 1 for further reference.

### 2.2. Study Selection

The selection of studies for this review occurred in a two-stage process. Initially, four independent reviewers (BDG, EP, RM, FR) assessed the titles and abstracts of all the retrieved papers. In the subsequent stage, these same reviewers individually examined the full texts of the papers identified in the first phase. Any discrepancies between the four reviewers were resolved by involving a senior researcher.

### 2.3. Quality Assessment

Given the heterogeneity of the included studies, the Grading of Recommendations, Assessment, Development, and Evaluation (GRADE) approach was employed to assess the quality of the evidence. This assessment was conducted by two reviewers (BDG and FR), with any disagreements resolved through discussion with an additional reviewer (AV). The GRADE standards categorize the quality of evidence as “high,” “moderate,” “low,” or “very low.” A “high-quality” rating suggests that future research is very unlikely to alter the existing evidence and that the true effect closely matches the estimated effect.

### 2.4. Data Extraction and Data Synthesis

Four independent researchers (SL, VM, AM, and FM) carried out data extraction for each included study, utilizing a standardized data extraction sheet in Microsoft Excel. The focus of this extraction encompassed several key subjects, including study design, participant characteristics, diagnosis of FEP, and digital device details derived from the original research. A meta-analysis was not conducted due to significant heterogeneity in study designs, interventions, outcome measures, and durations. Therefore, a narrative synthesis was employed to summarize the findings systematically.

### 2.5. Risk of Bias Assessment

Two independent researchers (VDS and MDA) assessed the quality of evidence from the eligible studies. The Revised Cochrane Risk of Bias Tool (RoB 2.0) was used to evaluate the risk of bias in randomized controlled trials (RCTs) (Table 2) [28].

## 3. Results

The final selection included 10 studies, comprising randomized controlled trials (RCTs), cohort studies, and open-label feasibility studies. Most studies utilized smartphone applications (e.g., ReMindCare, My Journey 3, Actissist, and Horyzons) aimed at symptom monitoring, medication adherence, and peer support. Other interventions included telepsychiatry platforms, digital Cognitive Behavioral Therapy for insomnia (CBT-I), and wearable devices for passive symptom tracking. Study durations varied, ranging from short-term interventions of 8–12 weeks to longer-term programs extending up to 18 months. Participants primarily included individuals diagnosed with FEP, with study objectives focused on improving functional recovery, engagement, and treatment adherence.

The results were categorized into three main themes—Clinician Perspectives on Telepsychiatry, Digital Mental Health Interventions, and Study Protocols—based on their relevance to rehabilitation-focused interventions and the breadth of available evidence on FEP management.

### 3.1. Clinician Perspectives on Telepsychiatry

Lal et al. [24] conducted a comprehensive examination of clinician experiences with telepsychiatry during the COVID-19 pandemic, revealing overwhelmingly positive feedback from mental health professionals who adopted this innovative approach. Clinicians reported significant improvements in patient engagement, streamlined follow-up processes, and enhanced continuity of care facilitated by platforms like REACTS. A standout finding was the dramatic increase in patient engagement through telepsychiatry, enabling more frequent interactions and effectively overcoming the logistical barriers often associated with traditional face-to-face appointments. This increased accessibility proved particularly beneficial for young adults experiencing their FEP, as it provided a comfortable and familiar environment conducive to encouraging open dialogue. Moreover, telepsychiatry played a vital role in maintaining continuity of care, which is crucial for individuals dealing with FEP, by allowing timely check-ins and necessary adjustments to treatment plans without the typical delays of scheduling in-person visits. Initially, technical challenges such as connectivity issues were encountered; however, these diminished over time as both clinicians and patients became more proficient with the technology, supported by targeted training and ongoing assistance. The findings of Lal et al. [24] underscore telepsychiatry not only its emergent role in emergency response but also its potential sustainability as an integral component of mental health services, significantly advancing accessibility and patient engagement.

### 3.2. Digital Mental Health Interventions

Lal et al. [26] examined Horyzons-Canada, a digital intervention to support young adults recovering from FEP. This program promotes peer support and coping strategies, crucial for recovery in this demographic. While participants valued these peer interactions, they also suggested improvements in onboarding and tailored content, which could enhance user experience and platform effectiveness. Steare et al. [25] investigated “My Journey 3”, a mobile app for self-management for FEP individuals. Despite positive recruitment and retention, low engagement highlighted barriers to integration into daily routines. Factors such as interface design and user motivation were identified as areas for improvement. Similarly, the Momentum trial [27] showed the potential for smartphone applications to enhance patient activation, though challenges in user adaptation and support systems indicated a need for more robust integration strategies. Exploring the economic angle, Engel [23] affirmed Horyzons’ cost-effectiveness and its benefits for social functioning post-discharge. Cella et al. [21] discussed combining active and passive symptom monitoring, showing promise for early warning systems. Conversely, Bucci et al. [20] found Actissist less effective than symptom-monitoring, underscoring the need for the continual refinement of digital interventions to enhance their effectiveness and relevance. Overall, these studies underscore both the potential and challenges of digital interventions, particularly in young adults recovering from FEP, emphasizing the need for enhanced user engagement strategies and evaluations of cost-effectiveness to shape future digital health solutions.

### 3.3. Study Protocols

Bonet et al. [19] evaluated ReMindCare, a smartphone app designed to improve care quality for FEP individuals. The study aimed to assess the app’s impact on medication adherence and early relapse detection, using daily and weekly assessments within a First Episode Psychosis Program (FEPP). The naturalistic, prospective design of the study, ongoing at a hospital in Valencia, Spain, seeks to provide longitudinal data over five years. Fowler et al. [22] outlined a study for a Digital Medicine System (DMS) integrating various components to enhance medication adherence and well-being in schizophrenia and FEP patients. The study’s innovative co-production methodology involved patients, caregivers, and healthcare professionals, aiming to maximize the system’s effectiveness in real-world settings. Beattie et al. [18] proposed a feasibility trial for digital Cognitive-Behavioural Therapy for Insomnia (CBT-I) within Early Intervention in Psychosis services. This study emphasizes evaluating the feasibility of Sleepio, a digital CBT-I program, to address insomnia prevalent in individuals experiencing first-episode psychosis.

To enhance the accessibility of the findings, an infographic summarizing the key interventions, outcomes, barriers, and study durations is provided below (Table 3). This visual representation aims to facilitate understanding of the study’s scope and implications for clinical practice and future research.

## 4. Discussion

The existing literature reflects a significant and transformative shift toward the adoption of digital health solutions in the management of FEP care [4]. These advancements have been associated with marked improvements in patient engagement and accessibility, fundamentally altering how mental health services are delivered. This review specifically focuses on rehabilitation-oriented digital interventions, addressing a critical yet often underexplored phase of FEP care. Rehabilitation, which aims to restore functional abilities, improve social integration, and foster long-term recovery, is a key phase in the treatment of FEP. The findings underscore the role of digital tools, such as smartphone applications, telepsychiatry platforms, and digital CBT, in supporting treatment adherence, facilitating symptom self-management, and enhancing patient engagement during the rehabilitation process. By targeting functional recovery and promoting self-efficacy, these interventions help address the multifaceted needs of individuals recovering from FEP. This focus on rehabilitation is particularly relevant as it bridges the gap between acute symptom stabilization and long-term recovery, ensuring sustained improvements in quality of life and functional outcomes. Digital interventions have also opened new avenues for reaching individuals who might otherwise face barriers to in-person treatment, creating opportunities for enhanced interaction and support [18]. Despite these positive developments, several challenges persist that could hinder the full potential of digital health solutions. Low user engagement remains a critical issue, often attributed to user interface design complexities, insufficient educational resources, and a lack of motivation among users to engage consistently with the digital tools provided [29]. Technical difficulties can also create significant barriers, with issues such as connectivity problems, software glitches, and platform usability often dissuading users from fully utilizing the services. Telepsychiatry, in particular, has emerged as a credible alternative to traditional face-to-face consultations, especially during times when in-person visits may not be feasible or safe [1]. The success of telepsychiatry hinges on adequate support and training for clinicians and patients alike. With comprehensive training programs in place, both mental health professionals and patients can develop the necessary skills to navigate digital platforms confidently [9]. This has resulted in a smoother transition to telehealth models, enabling mental health providers to maintain continuity of care in the face of unprecedented challenges like the COVID-19 pandemic [24]. Digital interventions, such as Horyzons-Canada and telepsychiatry, hold significant potential to revolutionize the clinical management of FEP by enhancing accessibility, personalization, and proactive care. These technologies facilitate tailored interventions through real-time data collection, enabling clinicians to adjust treatment plans dynamically based on symptom monitoring and user feedback. Moreover, predictive analytics integrated into digital tools can detect early signs of relapse, allowing timely interventions that reduce the risk of hospitalization and improve recovery trajectories. DMS and smartphone applications support treatment adherence by providing precise monitoring of medication intake and incorporating behavioral prompts to enhance patient compliance. These tools can be seamlessly integrated into hybrid care models, complementing traditional in-person services with digital platforms to create comprehensive and scalable care pathways. By reducing geographic and systemic barriers, digital technologies extend high-quality care to underserved populations, while structured programs, such as digital Cognitive Behavioral Therapy (CBT), address co-occurring conditions like insomnia or anxiety, further enhancing therapeutic outcomes. To address the barriers related to technological literacy, digital interventions should incorporate user-friendly interfaces that simplify navigation and interaction. Examples include incorporating visual cues, step-by-step tutorials, and multilingual support to make platforms accessible for users with varying levels of digital proficiency. Platforms like Horyzons-Canada have demonstrated the utility of intuitive design features, which reduce dropout rates and improve engagement. Additionally, passive monitoring using wearables and smartphones generates longitudinal data, offering clinicians deeper insights into subtle behavioral or physiological changes, which can inform nuanced clinical decisions. These advancements also promote economic benefits, including reduced healthcare costs and greater scalability, making them a valuable addition to overburdened mental health systems. By empowering patients with self-management tools, peer support networks, and psychoeducational resources, digital interventions foster autonomy and community engagement, critical factors in FEP recovery. Future developments should focus on refining these tools through longitudinal trials, demographic customization, and integration with genomic and neurobiological data, ensuring their efficacy and relevance in diverse clinical contexts. Further details on emerging technologies, including AI-driven tools for symptom monitoring and predictive analytics, are provided in Appendix A. Such innovations have the potential to redefine the delivery of mental health care, transforming it into a more inclusive, efficient, and patient-centered paradigm [23]. However, to fully realize these economic benefits, healthcare organizations must pay close attention to several critical components. User onboarding must be prioritized to ensure that users feel comfortable and competent in their use of digital platforms. This includes providing comprehensive training materials, user guides, and tutorials that are easily accessible for both clinicians and patients [25,26]. Personalization is critical for maximizing the effectiveness of digital interventions. AI-powered tools can analyze user data to provide tailored therapeutic recommendations and adapt content to the specific needs of patients. For instance, symptom trackers and feedback systems can adjust dynamically based on individual progress, ensuring that the intervention remains relevant and engaging over time. Additionally, the educational content provided through these digital interventions should be robust and tailored to meet the diverse learning needs of users, allowing for personalized learning experiences that cater to various levels of technological familiarity. Peer support mechanisms are also crucial for enhancing user satisfaction and engagement [3]. Creating opportunities for users to connect with peers can encourage a sense of community and belonging, which is particularly essential for individuals navigating the challenges of mental health recovery. Digital interventions that facilitate structured peer interactions, moderated support groups, and community-building activities can significantly enhance the overall user experience and improve retention rates. This review has several limitations. First, the heterogeneity of study designs and outcome measures limit our ability to perform a meta-analysis. Second, the risk of publication bias cannot be ruled out. Third, most studies had small sample sizes and short durations, which may limit generalizability. The exclusion of non-English studies may have led to the omission of relevant findings. While this decision ensures a rigorous evaluation process, it may limit the generalizability of the findings to non-English-speaking populations. Future reviews should consider including multilingual research teams to address this limitation and ensure broader applicability. Lastly, the heterogeneity in study designs, including differences in methodologies, participant characteristics, and intervention durations, posed challenges for the narrative synthesis. While this variability allowed for a broader perspective on digital interventions for first-episode psychosis, it also limited the ability to draw definitive conclusions and identify consistent patterns across studies. This heterogeneity may have introduced variability in outcomes, reducing the precision of the overall synthesis. Future research should aim to standardize methodologies and outcome measures to improve comparability and strengthen the evidence base.

## 5. Conclusions

In conclusion, the integration of digital health solutions in the care of FEP represents a significant advancement in mental health treatment, offering the potential for improved patient outcomes and greater engagement. However, to fully capitalize on these benefits, it is crucial to address challenges related to user engagement and technical issues. Success will depend on strategies such as enhancing user onboarding, offering comprehensive educational resources, and building robust peer support networks to foster a sense of community and connection among users. By focusing on user-centered design and ensuring continuous platform updates, healthcare providers can maintain the relevance and effectiveness of digital interventions for diverse populations. Prioritizing these factors will not only help meet the current needs of patients but also enable healthcare systems to adapt to the changing landscape of mental healthcare, promoting a more inclusive and responsive treatment environment. Moreover, adapting interventions for different cultural contexts is essential to ensure inclusivity and effectiveness. This includes offering culturally relevant content, translation into multiple languages, and considering local beliefs and values when designing interventions. We urge funding agencies to prioritize investments in scalable digital health technologies. By allocating resources to the development accessible, cost-effective platforms, agencies can significantly enhance the reach of mental health interventions, particularly in underserved communities.

Collaborating with community leaders and stakeholders can further enhance cultural appropriateness and acceptance.

## 6. Future Directions

Future research in digital interventions for FEP should prioritize longitudinal studies to assess the long-term impact of these tools on recovery trajectories. Understanding how digital interventions influence sustained recovery over time is crucial for refining treatment protocols and ensuring that these solutions effectively support individuals throughout their mental health journeys. Longitudinal research can reveal patterns of engagement, resilience, and outcomes, providing valuable insights into how digital platforms can be optimized for ongoing recovery beyond initial engagement [30]. Longitudinal research is necessary to evaluate the long-term efficacy and impact of digital interventions on recovery trajectories. Tracking patient outcomes over extended periods can help identify factors that contribute to sustained recovery and refine intervention strategies accordingly. Additionally, research should employ mixed-methods designs, combining both quantitative metrics and qualitative feedback. This approach will offer a more comprehensive understanding of user experiences and intervention effectiveness. Quantitative data can measure adherence, symptom reduction, and satisfaction, while qualitative data can capture personal narratives that highlight the emotional and contextual factors affecting user engagement. These insights will be essential for identifying barriers to effective use of these technologies and refining digital interventions to better address user needs [31]. The integration of advanced predictive analytics, including machine learning techniques, holds great potential for improving monitoring systems in digital mental health interventions. By analyzing user data, these systems could detect early signs of relapse or deterioration, enabling proactive care strategies tailored to individual needs. This shift towards personalized care emphasizes early intervention rather than reactive responses, potentially preventing crises before they escalate [1].

Future research should also focus on refining scalable, user-friendly digital platforms. As mental health care integrates digital solutions, it is vital to assess the accessibility and usability of these interventions across diverse populations. Engaging users in ongoing feedback loops will ensure platforms are both technically effective and culturally relevant. Research should explore how digital interventions impact different demographic and clinical subgroups, accounting for factors such as age, socioeconomic status, and co-occurring conditions. Tailoring interventions to diverse user profiles will enhance their effectiveness and increase engagement [32]. AI and machine learning offer significant potential for enhancing digital interventions. Predictive analytics can identify early warning signs of relapse or non-adherence, enabling proactive intervention. For example, machine learning models integrated into tools like ReMindCare can process real-time data to generate actionable insights for clinicians.

Finally, collaboration between researchers, clinicians, and technology developers is essential for the success of digital interventions for FEP. Partnerships can bridge the gap between research findings and clinical applications, ensuring that digital health solutions are based on the latest evidence and aligned with clinical practices [33]. This interdisciplinary collaboration will help create comprehensive, integrated care pathways that combine the strengths of both traditional and digital therapeutic approaches [34]. Clinicians can integrate digital tools into hybrid care models to complement traditional in-person services. For example, smartphone applications like ReMindCare can monitor symptoms and adherence in real-time, providing actionable insights between face-to-face sessions. Similarly, digital platforms such as Horyzons-Canada can foster peer support and therapeutic engagement, addressing gaps in accessibility and continuity of care.

In summary, the future of research in digital interventions for FEP should focus on longitudinal studies, mixed-methods designs, predictive analytics, and collaborative frameworks. By prioritizing these areas, researchers can deepen the understanding of technology’s role in mental health care, improving patient outcomes and enhancing the overall quality of care for individuals experiencing first-episode psychosis. With continued innovation, adaptation, and user-centric design, digital health solutions can become integral to effective recovery strategies (Table 4).

## Figures and Tables

**Figure 1 brainsci-15-00080-f001:**
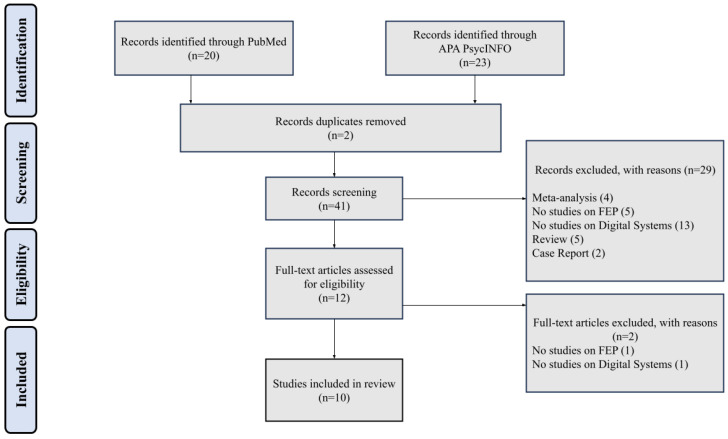
PRISMA search process.

**Table 1 brainsci-15-00080-t001:** Studies using digital devices in subjects with FEP.

Author	Year and Country	Period	Study Design	Study Sample	Treatment	Comparator Group	Measures	Outcome
Beattie & Robb [18]	2023, UK	Not specified	Feasibility study protocol Grade: **	N = 40; mean age not specified; first-episode psychosis with insomnia; capacity to consent and access to technology required.	Sleepio digital CBT program; 6 sessions (15–20 min each); cognitive, behavioral, and educational techniques delivered via animated avatar; accessible via web and iOS applications.	None	Changes in SCI-2, ISI, DASS-21, R-GPTS, SPEQ, and Fear of COVID-19 scores.	Digital CBT interventions, like Sleepio, show promise in improving insomnia and mental health in first-episode psychosis. Feasibility metrics highlight potential for integration despite barriers like digital poverty, with qualitative insights informing future scalability.
Bonet & Torous [19]	2020, Spain and US (Massachusetts)	50 weeks, with a minimum of 18 days and a maximum of 594 days	Observational Study Grade: **	N = 57 (gender distribution not specified); aged between 17–65 years old; DSM-5 diagnosis of psychotic disorder; first-episode psychosis patients enrolled in a specialized program.	ReMindCare app; 3 daily assessment of anxiety, sadness, and irritability; weekly questionnaires on medication adherence, side effects, and prodromal symptoms; integrated with the hospital electronic medical record; available for a maximum period of 5 years.	None	Changes in CGI, GAF, PANSS, PAS, DAI-10, BCIS, sociodemographic information, clinical information and SMAQ. Number of relapses, number of visits to urgent care units at the hospital, number of hospital admissions	The ReMindCare app aims to improve treatment delivery, enhance patient-clinician communication, increase adherence to medication, and facilitate early relapse detection. Anticipated outcomes include reduced hospital admissions, fewer urgent care visits, and improved insight and alliance between patients and clinicians. Early findings suggest high retention and compliance, supporting its integration into routine psychiatric care.
Bucci & Berry [20]	2024, UK	12 weeks	Randomized controlled trial (RTC) Grade: ****	N = 172 (108 M, 64 F); mean age 29.1 ± 9.2; ICD-10 diagnosis of schizophrenia-spectrum psychosis; within 5 years of first-episode onset, with moderate symptom severity.	Actissist app based on CBT principles; 12-week treatment window; daily question-answer exchanges addressing relapse risk factors; integrated multimedia tools for coping and behavior strategies; the control group received ClinTouch symptom monitoring app.	Comparison group used a monitoring app called “ClinTouch”	Changes in PANSS, PSYRATS, CDSS, PSP, PCPW, QPR, WEMWEBS, ISMI, ERS, AUDIT	No significant difference in PANSS total score between Actissist and ClinTouch groups; both groups showed symptom improvement over time; Actissist was found safe and acceptable but not superior to symptom monitoring; secondary measures also showed no significant differences.
Cella & He [21]	2019, UK	1 week and 2 days	Pilot Study Grade: ***	N = 15 (12 M, 3 F); mean age 28.1 ± 3.8; first-episode psychosis; onset within the last 12 months.	ClinTouch mobile app for active symptom monitoring; Empatica E4 wearable device for passive monitoring (EDA and HRV); 10-day simultaneous use; daily app prompts assessing psychotic symptoms and distress levels.	None	Changes in PANSS, GAF, EDA, HRV, Motion and Skin Temperature	EDA levels significantly higher during distressing hallucinations and delusions; no significant changes in HRV during distressing symptoms; high acceptability and compliance with app and wearable device; study supports the feasibility of combining active and passive monitoring for early psychosis.
Fowler & Cope [22]	2019, US (New York)	8 weeks	Study Protocol Grade: **	N = 60 (gender distribution not specified); aged 18–65 years; diagnosis of schizophrenia, schizoaffective disorder, or first-episode psychosis (ICD-10 codes F20/F25); prescribed oral atypical antipsychotics (aripiprazole, olanzapine, quetiapine, or risperidone).	Digital Medicine System (DMS); composed of a sensor-embedded oral tablet, wearable patch, mobile application, and web-based dashboard; 8-week intervention with continuous patch wear and ingestion monitoring; data accessible to patients, caregivers (with consent), and HCPs via secure web portals.	None	Changes in PANSS, subjects usability and satisfaction scale, physician utility survey, caregiver/support person involvement scale, personal and social performance scale, patient activation measure-mental health scale.	High acceptance and engagement reported among patients and HCPs; preliminary findings suggest feasibility of DMS for monitoring adherence and supporting clinical decisions; no formal power calculations as this was an exploratory feasibility study.
Engel & Alvarez-Jimenez [23]	2024, Australia	18 months (approx. 78 weeks)	Randomized Controlled Trial (RCT) Grade: ****	N = 170 (90 M, 80 F); mean age 20.91 ± 2.88; young adults diagnosed with first-episode psychosis; in remission and nearing discharge from specialized services.	Horyzons online intervention; 18-month duration; based on the Moderated Online Social Therapy model (MOST); which integrates: online therapy, peer-to-peer social networking, and clinician support; targeting social functioning, vocational recovery, and relapse prevention.	Treatment as Usual (TAU)	Changes in PSP, AQoL-8D, QALYs	Horyzons group demonstrated significantly lower costs compared to treatment as usual (TAU); small improvements in social functioning (PSP) over TAU, though not statistically significant; greater engagement with therapeutic components linked to improved outcomes; Horyzons shown to be cost-effective, offering potential budget savings and supporting long-term recovery.
Lal & Abdel-Baki [24]	2023, Canada	Not specified	Qualitative Study Grade: *	N = 26 (22 F, 4 M); mental health service providers working in SEI for psychosis program; professional backgrounds included social workers, nurses, physicians, occupational therapists, and peer support workers; conducted during COVID-19 using telepsychiatry services (REACTS platform).	Telepsychiatry using REACTS platform; video/audio communication, file sharing, and messaging capabilities; used for patient engagement, therapy sessions, follow-ups, and family meetings; clinician training provided pre-pandemic, with additional booster training during the pandemic.	None	Primary: Frequency and duration of telepsychiatry sessions using the REACTS platform. Secondary: Clinician-reported safety, ease of use, and satisfaction with telepsychiatry; technical issues encountered (e.g., sound, image quality, session initiation); perceptions of client engagement, continuity of care, and technical support needs; qualitative analysis of barriers and facilitators to telepsychiatry adoption over time.	Positive perceptions of telepsychiatry maintained over time; improvements in perceived ease of use and session efficiency; benefits included enhanced client engagement and continuity of care; barriers included initial usability challenges and technical issues; clinicians supported continued use of telepsychiatry post-pandemic.
Steare & O’Hanlon [25]	2020, UK	Median of 38.1 weeks	Randomized Controlled Trial (RCT) Grade: ****	N = 40 (28 M, 12 F); mean age 29.7 ± 9.78; ICD-10 diagnosis of psychotic disorders (F20–F29) and mood disorders (F30–F39); users of Early Intervention in Psychosis (EIP) services owning Android smartphones.	My Journey 3 app; features include symptom tracking, relapse prevention, medication reminders, and psychoeducation; delivered with clinician support for set-up and training; median access duration 38.1 weeks; the control group received treatment as usual (TAU).	Treatment as Usual (TAU)	Changes in QPR, WEMWBS, DIALOG, PANSS, SES, SIX, qualitative survey	Feasibility confirmed with 75% 12-month retention; My Journey 3 use was low (median use 16.5 times per participant); no significant differences in primary or secondary outcomes between groups; qualitative feedback indicated acceptability but barriers included technical issues and limited clinician support; relapse rates were low across both groups.
Lal &Tobin [26]	2023, Canada	8 weeks	Qualitative focus group study Grade: *	N = 23 (6 F, 2 M, 1 non-binary, 14 gender not specified); aged 18–34 years (mean age 26.9 ± 6.2); diagnosed with psychotic disorders within the first three years of treatment; recruited from a Canadian first-episode psychosis program.	Horyzons-Canada (HoryzonsCa) online intervention; platform included peer networking, therapeutic content, and clinical moderation; focused on stress management, self-improvement, and recovery strategies; pilot study duration was 8 weeks.	None	Participant feedback on recovery strategies and platform usability; focus group data exploring core themes: ease of use, peer networking, therapeutic content, and moderation; engagement metrics from platform use (logins and activity completion); qualitative thematic analysis.	Participants found HoryzonsCa helpful for recovery and stress management; key strengths included ease of navigation, connection with peers, and access to therapeutic content; challenges included limited awareness of platform features and need for deeper, personalized content; positive feedback on moderator support and desire for expanded platform scale and functionality.
Vitger & Austin [27]	2019, Denmark	6 months, with data collection at baseline, 3 months and 6 months	Randomized Controlled Trial (RCT) Grade: ****	N = 260 (gender distribution not specified); aged 18+; diagnosed with schizophrenia, schizotypal, or delusional disorders (ICD-10 codes F20–F29); receiving treatment in OPUS specialized early intervention program; participants must own a smartphone and have daily access.	Momentum smartphone app with TAU; features include daily self-evaluation (stress, sleep, well-being), recovery goal setting, and action plans; web portal for clinicians to view patient input before consultations; control group received TAU only; 6-month intervention.	Treatment as Usual (TAU)	Changes in CHAI-MH, CDMS, WAI (short form), ASH, GSE, PEPPI-5, PrepDM, CSQ-8, ARQ, sociodemographic measures, SES, PSP, GAF, SAPS, SANS	Expected: Improved patient activation and preparedness for SDM; potential secondary benefits include enhanced alliance, functioning, self-efficacy, and satisfaction; data on app engagement correlated with improvements; study aims to validate digital tools for supporting SDM in outpatient mental health care.

Note: CBT: Cognitive Behavioral Therapy; DSM-5: Diagnostic and Statistical Manual of Mental Disorders, Fifth Edition; RTC: Randomized Controlled Trial; ICD-10: International Classification of Diseases, Tenth Revision; DMS: Digital Medicine System; HCPs: Healthcare Providers; TAU: Treatment As Usual; PSP: Personal and Social Performance Scale; OPUS: OPUS Early Intervention Program; SDM: Shared Decision-Making; SCID-2: Structured Clinical Interview for DSM Disorders (Second Version); ISI: Insomnia Severity Index; DASS-21: Depression Anxiety Stress Scales—21 Items; R-GPTS: Revised Green Paranoid Thoughts Scale; SPEQ: Specific Psychotic Experiences Questionnaire; Fear of COVID-19 Scale; CGI: Clinical Global Impression; GAF: Global Assessment of Functioning; PANSS: Positive and Negative Syndrome Scale; PSYRATS: Psychotic Symptom Rating Scales; CDSS: Calgary Depression Scale for Schizophrenia; PSP: Personal and Social Performance Scale; PCPW: Perceived Criticism and Perceived Warmth Scale; QPR: Questionnaire about the Process of Recovery; WEMWBS: Warwick-Edinburgh Mental Well-being Scale; ISMI: Internalized Stigma of Mental Illness Inventory; ERS: Empowerment Rating Scale; AUDIT: Alcohol Use Disorders Identification Test; EQ-5D-5L: EuroQol Five-Dimensional Questionnaire (5-Level Version); CSRI: Client Service Receipt Inventory; SES: Service Engagement Scale; SIX: Social Inclusion Index; MHCS: Mental Health Confidence Scale; CHAI-MH: Consumer Health Activation Index for Mental Health; PrepDM: Preparedness for Decision-Making Scale; WAI-S: Working Alliance Inventory—Short Form; ASH: Adult State Hope Scale; GSE: General Self-Efficacy Scale; ARQ: App Rating Questionnaire; SAPS: Scale for the Assessment of Positive Symptoms; SANS: Scale for the Assessment of Negative Symptoms; DAI-10: Drug Attitude Inventory—10 Items; BCIS: Beck Cognitive Insight Scale; SMAQ: Simplified Medication Adherence Questionnaire; QFQ: Quantitative Feedback Questionnaire (27 items); AQoL-8D: Assessment of Quality of Life (8 Dimensions); CUDIT-R: Cannabis Use Disorders Identification Test—Revised; DUDIT: Drug Use Disorder Identification Test; ASSIST: Alcohol, Smoking, and Substance Involvement Screening Test; TLFB: Timeline Followback (for drugs and alcohol use); PAS: Premorbid Adjustment Scale; EDA:Electrodermal activity; HRV:heart rate variability; REACTS Remote Education, Augmented Communication, Training, and Supervision; RGPTS Revised Green Paranoid Thoughts Scale, MOST Moderated Online Social Therapy, QALYs Quality-Adjusted Life Years, DIALOG, EIP Early Intervention in Psychosis, CDMS, PEPPI-5 Perceived Efficacy in Patient-Physician Interaction—5 items, CSQ-8 Client Satisfaction Questionnaire. GRADE: * very low; ** low; *** moderate; **** high.

**Table 2 brainsci-15-00080-t002:** Evaluation of the risk of bias for Randomized Studies using the RoB 2.0 Tool.

References	Name of Digital Health Interventions	Overall Risk	Randomization	Intervention	Missing Data	Outcome Measurement	Grade
Bucci & Berry (2024) [20]	Actissist	+/−	-	+/−	-	+/−	+/−
Steare & O’Hanlon (2020) [25]	ARIES	+/−	-	+/−	+/−	+/−	+/−
Engel & Alvarez-Jimenez (2024) [23]	Horyzons	-	-	-	-	-	+
Vitger & Austin (2019) [27]	Momentum	+/−	-	+/−	-	+/−	+/−

**Table 3 brainsci-15-00080-t003:** Overview of digital interventions for first-episode psychosis: This table summarizes key interventions, their outcomes, identified gaps, and study durations. It highlights the potential of digital tools, such as smartphone applications and telepsychiatry, in enhancing engagement, supporting recovery, and improving adherence while addressing barriers like low user engagement and technical challenges.

Author/Year	Intervention	Key Outcomes	Gaps/Barriers	Study Duration
Lal et al., (2023) [24]	Horyzons Platform	Improved social functioning; cost-effective	Need for tailored content and better onboarding	8 weeks
Steare & O’Hanlon (2020) [25]	My Journey 3 App	Feasibility confirmed; barriers to daily use	Low engagement and interface issues	38.1 weeks (median)
Engel & Alvarez-Jimenez (2024) [23]	Horyzons Online Intervention	Cost-effective; moderate engagement improvement	Limited social functioning improvement	18 months
Bucci & Berry (2024) [20]	Actissist App	Safe but not superior to symptom monitoring	Requires further refinement for efficacy	12 weeks
Cella et al. (2019) [21]	ClinTouch & Wearable Device	High acceptability; early warning feasibility	Technical issues; need larger sample size	10 days
Beattie & Robb (2023) [18]	Digital CBT for Insomnia (Sleepio)	Improved insomnia; highlights scalability challenges	Digital poverty and onboarding challenges	6 sessions over 6 weeks
Bonet & Torous (2020) [19]	ReMindCare App	Enhanced adherence and early relapse detection	Technical challenges; need for long-term data	50 weeks
Fowler & Cope (2019) [22]	Digital Medicine System (DMS)	High acceptance; feasible for adherence monitoring	Exploratory feasibility study; lacks power	8 weeks
Vitger & Austin (2019) [27]	Momentum App	Improved patient activation and preparedness	User adaptation and support system challenges	6 months
Lal & Tobin (2023) [26]	Horyzons-Canada (HoryzonsCa)	Helpful for recovery and stress management	Limited awareness of platform features	8 weeks

**Table 4 brainsci-15-00080-t004:** Digital interventions: potential and limitations in FEP management. This summary captures the essence of the reviewed findings and highlights the potential and limitations of digital interventions in addressing the unique challenges of FEP management.

Category	Key Findings
Purpose of Digital Interventions	- Enhance patient engagement, reduce stigma, and improve access to mental health services.
	- Aid early diagnosis and intervention to prevent long-term negative outcomes.
Types of Interventions	- Telepsychiatry: Effective for continuity of care, especially during COVID-19.
	- Smartphone Apps: Tools like ReMindCare and Horyzons for symptom tracking, relapse prediction, and peer support.
Benefits	- Timely access to care and increased frequency of patient-provider interactions.
	- Cost-effectiveness and improved treatment adherence.
	- Potential to integrate active/passive monitoring tools.
Challenges	- Low user engagement due to complex interfaces and lack of motivation.
	- Technical issues (e.g., connectivity, usability).
	- Need for tailored, culturally relevant interventions.
Notable Applications	- Horyzons-Canada: Peer support for recovery; emphasizes engagement improvements.
	- ReMindCare: Tracks medication adherence and early relapse signs.
	- CBT-I Programs: Digital therapy for insomnia co-occurring with FEP.
Implementation Strategies	- Importance of training for clinicians and patients.
	- Robust onboarding and user education.
	- Incorporation of peer support and personalized features.
Research Insights	- Longitudinal studies and mixed-methods approaches are vital.
	- Predictive analytics and machine learning for early relapse detection.
	- Integration with traditional care pathways enhances outcomes.
Future Directions	- Develop scalable, user-friendly platforms.
	- Focus on demographic diversity in intervention design.
	- Collaborate among clinicians, researchers, and developers to optimize solutions.

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
