# Peer review of "Digital Interventions for the Rehabilitation of First-Episode Psychosis: An Integrated Perspective"

_brainsci, 2025, doi:10.3390/brainsci15010080_

Round 1
Reviewer 1 Report (Previous Reviewer 2)
Comments and Suggestions for Authors
Dear authors
Thank you very much for the revision of the manuscript and the comprehensive consideration of the comments. In my opinion, the changes have helped to improve the manuscript and I have no further comments.
Author Response
Dear Reviewer 1,
Thank you for your thoughtful and comprehensive review of our manuscript. We greatly appreciate your detailed feedback, and your acknowledgment of the improvements made. We are pleased that the revisions have enhanced the manuscript and that you have no further comments. Your input has been invaluable in refining our work.
Thank you once again for your time and effort in reviewing our manuscript.
Best regards,
Francesco Monaco ad Ernesta Panarello
On behalf of all co-authors

Reviewer 2 Report (Previous Reviewer 1)
Comments and Suggestions for Authors
As noted in my initial review, the manuscript reports findings from a narrative review of the use of digital interventions for rehabilitation for individuals with first-episode psychosis (FEP). The authors are commended for providing a thorough response to my suggestions and a clear summary of the revisions. In particular, I appreciate the authors’ effort to clarify formulation of the research question, the protocol of the literature search and presentation of the results. I just spotted that the extra table on p. 17 -18 needs to be signposted in the main text for reader’s easy reference. I have no further suggestions.
Author Response
Dear Reviewer,
Thank you very much for your detailed feedback and for your kind words regarding the revisions made to our manuscript. We are glad that the clarifications provided regarding the research question, literature search protocol, and results presentation were well received.
Regarding your observation about the extra table on pages 17–18, we would like to note that we have referenced Table 3 in the final section of the text as a summary of our work. Its purpose is to provide a concise overview of the findings and conclusions discussed throughout the manuscript. We hope this ensures ease of reference for readers.
Thank you once again for your thoughtful and constructive comments, which have helped us improve the manuscript.
Best regards,
Francesco Monaco ad Ernesta Panarello
On behalf of all co-authors

This manuscript is a resubmission of an earlier submission. The following is a list of the peer review reports and author responses from that submission.
Round 1
Reviewer 1 Report
Comments and Suggestions for Authors
The manuscript reports findings from a narrative review of the use of digital interventions in first-episode psychosis (FEP). The authors summarized examples of the use of digital devices for rehabilitation in individuals with FEP, providing an overview to researchers and clinicians on existing evidence of digital interventions for this specific population of individuals. I have some comments/ suggestions for the author’s consideration.
Introduction:
- The introduction is comprehensive and informative in providing an overview of digital intervention in first-episode psychosis, covering multiple aspects of digital intervention ‘to facilitate early symptom monitoring, improve access to care, and bring up more immediate communication between patients and healthcare professionals’ (lines 75 – 76, p. 2). However, the review only focuses on the rehabilitation aspects of digital intervention, according to the search term used. Therefore, the introduction would focus more on emphasizing the digital intervention specific to rehabilitation for the readers.
- I wonder about the scope of the review which only focused on “rehabilitation” of digital intervention. As mentioned in the Introduction, the potential of digital intervention is far-reached covering various aspects of healthcare. Meanwhile, there are other trials or interventions with a digital component used in individuals with FEP (such as Acceptance and Commitment Therapy combined with ecological momentary intervention, Vaessen et al., 2019), which should fall on the stage set in the Introduction. The relevance and importance of a synthesis on the use of digital devices in FEP ‘rehabilitation’ should be further crafted and justified in the Introduction and the Methods. The title should also reflect this conceptual formulation and methodological decision of the review.
Vaessen, T., Steinhart, H., Batink, T., Klippel, A., Van Nierop, M., Reininghaus, U., & Myin-Germeys, I. (2019). ACT in daily life in early psychosis: An ecological momentary intervention approach. Psychosis, 11(2), 93-104.
Methods:
- Please double-check the descriptions of databases used in the search, as there are inconsistencies throughout the manuscript: lines 129 – 130, p. 3, two major health-related databases: PubMed and APA PsycINFO; line 137, p. 3, Embase or Medline; Figure 1 indicated 3 sources: PubMed, APA, and PsychINFO (but APA PsycINFO should be considered one?)
- Is the review preregistered? If so, it would be good to mention that.
Results:
- The Results section would benefit from a brief descriptive overview of the included studies, such as types of digital interventions used and study design, based on information in Table 1.
Discussion:
- The Discussion should also highlight further the focus of rehabilitation in this review, which should be made more explicitly to the readers.
Other comments:
- The readability of the Introduction and Discussion could be improved by additional paragraphing, so the flow of logic between arguments would stand out more.
- How do the table at the end of the manuscript and Appendix 1 fit into the main text manuscript? Please signpost them in the main text for readers’ easy reference or delete them if they are irrelevant.
Reviewer 2 Report
Comments and Suggestions for Authors
Dear Authors,
Thank you for the opportunity to review the manuscript on this important topic. With regard to the title, it should be clear from the title that this is a systematic review, as recommended by PRISMA. The abstract fulfils the requirements, but the headings deviate from the recommendations of the PRISMA for Abstracts Checklist (e.g. Objectives instead of Background or Conclusions instead of Discussion). The introduction is very comprehensive and refers to current and relevant literature (the introduction refers to a total of 16 references, which is half of the total number of references used). A clear and comprehensible research question is not described. The description of the methods is comprehensible. Unfortunately, no statements are made on the evaluation of the included studies (e.g. with regard to risk of bias). Reasons should also be given as to why a meta-analysis is out of the question. The presentation of the flow diagram is comprehensible, but here too the authors deviate from the PRISMA recommendations (e.g. ‘Records after duplicates removed (n=2)’ - this is not comprehensible in this form). The presentation of the results as a ‘Relevant Section’ is not comprehensible in this form. Why were the three points (Clinician Perspectives on Telepsychiatry, Digital Mental Health Interventions and Study Protocols) selected and why not the categories reported in the appendix (pp. 15-16)? Where do the ‘most commonly used AI approaches in clinical practice’ presented in Appendix 1 come from? Are these also results from the review? Unfortunately, there is no reference to Table 1 (or to the appendices) in the presentation of results. The discussion is very general and not very specific with regard to the topic of ‘first episode psychosis’ (e.g. lines 226-238). The discussion repeatedly refers to studies that have already been used in the introduction (References 1, 3, 4, 9). The Conclusion and the very extensive “Future Direction” also do not correspond to the PRISMA recommendations (PRISMA: Discussion including discussion of limitations). Unfortunately, the limitations of the work are not discussed at any point. Overall, the authors should follow the PRISMA recommendations more closely (please complete the checklist for reviewers). Information should also be added as to why the review was not registered. The discussion should focus more on the results and, above all, describe the limitations of the methodological approach.
